# Mechanism of Resistance to Epidermal Growth Factor Receptor-Tyrosine Kinase Inhibitors and a Potential Treatment Strategy

**DOI:** 10.3390/cells7110212

**Published:** 2018-11-15

**Authors:** Tatsuya Nagano, Motoko Tachihara, Yoshihiro Nishimura

**Affiliations:** Division of Respiratory Medicine, Department of Internal Medicine, Kobe University Graduate School of Medicine, 7-5-1 Kusunoki-cho, Chuo-ku, Kobe 650-0017, Japan; mt0318@med.kobe-u.ac.jp (M.T.); nishiy@med.kobe-u.ac.jp (Y.N.)

**Keywords:** acquired resistance, T790M, C797S, alternative pathway, transformation, primary resistance

## Abstract

Treatment with epidermal growth factor receptor tyrosine kinase inhibitors (EGFR-TKIs) improves the overall survival of patients with *EGFR*-mutated non-small-cell lung cancer (NSCLC). First-generation EGFR-TKIs (e.g., gefitinib and erlotinib) or second-generation EGFR-TKIs (e.g., afatinib and dacomitinib) are effective for the treatment of *EGFR*-mutated NSCLC, especially in patients with *EGFR* exon 19 deletions or an exon 21 L858R mutation. However, almost all cases experience disease recurrence after 1 to 2 years due to acquired resistance. The *EGFR* T790M mutation in exon 20 is the most frequent alteration associated with the development of acquired resistance. Osimertinib—a third-generation EGFR-TKI—targets the T790M mutation and has demonstrated high efficacy against *EGFR*-mutated lung cancer. However, the development of acquired resistance to third-generation EGFR-TKI, involving the cysteine residue at codon 797 mutation, has been observed. Other mechanisms of acquired resistance include the activation of alternative pathways or downstream targets and histological transformation (i.e., epithelial–mesenchymal transition or conversion to small-cell lung cancer). Furthermore, the development of primary resistance through overexpression of the hepatocyte growth factor and suppression of Bcl-2-like protein 11 expression may lead to problems. In this report, we review these mechanisms and discuss therapeutic strategies to overcome resistance to EGFR-TKIs.

## 1. Introduction

The epidermal growth factor receptor (EGFR) is a receptor protein penetrating the cell membrane. The kinase domain is composed of the N and C lobes and an adenosine triphosphate (ATP) binding cleft between the two lobes. Following the binding of a ligand to the receptor, an asymmetric dimer is formed and the phosphate of ATP is transferred to the tyrosine residue of the regulatory domain. Various proteins are bound to this phosphorylated tyrosine and signals are transmitted downstream through the rat sarcoma (RAS)-rapidly accelerated fibrosarcoma (RAF)-mitogen-activated protein kinase (MAPK) and phosphatidylinositol 3-kinase (PI3K)-protein kinase B (PKB, also known as AKT) pathways. EGFR tyrosine kinase inhibitors (TKIs) competitively inhibit ATP with cleft of the kinase domain. Gefitinib, a first-generation EGFR-TKI, greatly improved the prognosis of *EGFR* gene mutation-positive patients with non-small-cell lung cancer (NSCLC) [1,2]. The vast majority (93%) of EGFR activation mutations (common mutations) occur in exons 19 to 21. In addition, deletion mutations in exon 19 and L858R mutations in exon 21 are particularly frequent (44.8% and 39.8%, respectively) [3]. Although EGFR-TKIs show an excellent therapeutic effect on *EGFR*-mutated NSCLC, most cancers develop resistance to EGFR-TKI. The T790M mutation in exon 20 is recognized in approximately half of EGFR-TKI tolerance cases [4,5]. In this review, we will summarize the mechanism of EGFR-TKI tolerance and present treatment strategies to overcome resistance. There are several outstanding reviews so far [6,7,8,9], but therapeutic drugs progress one after another and a new resistance mechanism emerges. Therefore, we would like to review focusing on the novel acquired resistance of the third-generation EGFR-TKI, osimertinib based on the latest academic information.

## 2. Acquired Resistance

### 2.1. T790M “Gatekeeper” Mutation

Despite the initial response to first-generation EGFR-TKIs, patients with NSCLCs harboring *EGFR* mutations acquire resistance to these agents, with a median time to disease progression of approximately 12 months [10,11,12,13,14]. The mechanisms of acquired resistance to first-generation EGFR-TKIs are summarized in Figure 1.

T790M was the first reported acquired resistance gene [4]. The T790M mutation structurally inhibits the binding of first-generation EGFR-TKIs to the ATP binding site. When the T790M mutation is added to the activation mutation of *EGFR*, the affinity of EGFR for ATP is increased, whereas the binding property of EGFR-TKIs is relatively decreased. Hence, the downstream signal is not inhibited and the cancer is tolerated. It has been thought that a small number of cancer cells having a secondary T790M mutation in addition to the active *EGFR* mutation may already be present prior to treatment with EGFR-TKIs and gradually become dominant during treatment with first-generation EGFR-TKIs (e.g., gefitinib and erlotinib). However, recent research revealed that T790M-positive cells also occur from initially T790M-negative single cells via genetic evolution [17].

Irreversible EGFR-TKIs were developed to overcome the T790M-mediated resistance. These second-generation EGFR-TKIs (e.g., afatinib and dacomitinib) irreversibly bind to the cysteine residue of the EGFR [18]. Although the second-generation EGFR-TKIs exert an effect on T790M, the half-maximal inhibitory concentration (IC_50_) is 30 to 100-fold higher than that observed with the activated mutation of *EGFR* [19]. Integrated analysis of two phase III studies (LUX-Lung 3 and 6 trials) showed that afatinib significantly prolonged overall survival compared with chemotherapy in patients with an activated *EGFR* mutation (hazard ratio, HR 0.81; 95% confidence interval, CI 0.66 to 0.99; *p* = 0.037; 27.3 vs. 24.3 months, respectively) [20,21]. Furthermore, a randomized phase IIb trial comparing afatinib and gefitinib in patients with NSCLC harboring an active *EGFR* mutation (LUX-Lung 7 trial) revealed superiority of afatinib regarding progression-free survival (PFS) (HR, 0.73; 95% CI, 0.57 to 0.95; *p* = 0.0073; 11.0 vs. 10.9 months, respectively) [22]. These results suggest that irreversible EGFR-TKIs delay the expression of T790M compared with first-generation EGFR-TKIs. However, the T790M mutation was detected in 36.4% or 47.6% of patients treated with afatinib [8,23]. In addition, the effect of afatinib was limited in patients with advanced NSCLC and T790M mutation who progressed during prior treatment with first-generation EGFR-TKIs [24].

The third-generation EGFR-TKIs (e.g., osimertinib) are pyrimidine-based irreversible EGFR-TKIs targeting the T790M mutation. Osimertinib irreversibly binds to the EGFR kinase by targeting the cysteine residue at codon 797 (C797) and has a highly selective inhibition activity [25]. It is effective for the activation of *EGFR*-mutated cell lines (IC_50_: 8–17 nmol/L in PC9 harboring an exon 19 deletion) and *EGFR* T790M-mutated cell lines (IC_50_: 5–11 nmol/L in H1975 harboring L858R/T790M). However, osimertinib is less effective in *EGFR* wild-type cell lines than the early-generation EGFR TKIs (IC_50_: 650 nmol/L in Calu3 and 461 nmol/L in H2073) [25,26,27,28]. The phase I/II clinical trials of osimertinib (AURA 1/AURA 2 trials) involving EGFR mutation-positive patients with NSCLC who became resistant to EGFR-TKI demonstrated that the median PFS was 9.6 months (95% CI, 8.3 to not reached), the overall response rate (ORR) of negative cases was 21% (95% CI, 12 to 34), and the median PFS was 2.8 months (95% CI, 2.1 to 4.3) [29]. A phase III trial confirming the effect of osimertinib in NSCLC patients with the *EGFR* T790M mutation (AURA3 trial) showed superiority of osimertinib over platinum therapy plus pemetrexed regarding PFS (HR, 0.30; 95% CI, 0.23 to 0.41; *p* < 0.001; 10.1 vs. 4.4 months, respectively) [30]. Intriguingly, another phase III trial confirming the effect of osimertinib in untreated NSCLC patients with an activated *EGFR* mutation (FLAURA trial) showed superiority of osimertinib over gefitinib or erlotinib regarding PFS (HR, 0.46; 95% CI, 0.37 to 0.57; *p* <0.001; 18.9 vs. 10.2 months, respectively) [31].

### 2.2. Acquired Resistance to Osimertinib

Mechanisms of acquired resistance to osimertinib are shown in Figure 2.

#### 2.2.1. C797S Mutation

Although osimertinib shows an excellent clinical effect in the activation of both the *EGFR* and *EGFR* T790M mutations, the development of resistance is inevitable [32,37,38,39]. The C797 is present in the ATP-binding pocket in which EGFR-TKIs bind irreversibly. Therefore, the point mutation of C797S in exon 20 of *EGFR*—a common mechanism of acquired resistance to osimertinib—induces resistance to third-generation EGFR-TKIs [18]. Indeed, C797S-mediated resistance third-generation EGFR-TKIs develops approximately within 1 year [32,40]. Intriguingly, previous clinical trials showed that the C797S is more likely to occur in *EGFR* exon 19 deletion mutation [32,38,39,41,42]. The localization of T790M and C797S on the allele may be helpful in determining the treatment strategy for C797S. If C797S and T790M are in trans (i.e., on separate alleles), the resistant cells may be sensitive to the combination of first- and third-generation EGFR-TKIs [43,44]. However, if C797S and T790M are in cis (i.e., on the same allele), the cells are resistant to all EGFR-TKIs [45]. The majority of *EGFR*-sensitizing mutations (i.e., exon 19 deletion, exon 20 insertion, and L858R/T790M *EGFR* mutants), except L858R, do not require asymmetric dimerization. Hence, they do not respond to treatment with cetuximab [46]. In contrast, the L858R/T790M/C797S which partly require dimerization moderately respond to treatment with cetuximab. To overcome the *EGFR*/T790M/C797S mutation, use of a potent anaplastic lymphoma kinase (ALK) inhibitor (i.e., brigatinib) or mutant selective allosteric inhibitor may be an option [47,48].

#### 2.2.2. Other *EGFR* Mutation

Other *EGFR* mutations include L792X mutation, G796S mutation, L718Q mutation and exon 20 insertion [49,50]. Exon 20 insertion inhibits the binding of EGFR-TKI to its binding site by adding residues at the N-lobe of *EGFR* [51]. Further studies are required for *EGFR* mutation, but cytotoxic anticancer agent is one of the treatment options for L718Q.

#### 2.2.3. Oncogenic Fusion

The latest research revealed a new resistance mechanism using plasma ctDNA genomic profile. Some of these novel mechanisms are oncogenic fusions, including fibroblast growth factor receptor 3 (*FGFR3*)-transforming acidic coiled coil-containing protein 3 (*TACC3*) gene fusion, neuropathic tyrosine kinase receptor type 1 (*NTRK1*)-tropomyosin 3 (*TPM3*) gene fusion, rearranged during transfection (*RET*)-ELKS-Rab6-interacting protein-CAST family member 1 (*ERC1*) gene fusion, and spectrin beta non-erythrocytic 1 (*SPTBN1*)-*ALK* gene fusion [50,52]. The *SPTBN1*-*ALK* fusion gene was formed by the fusion of exon 7 of the *SPTBN1* gene with exon 20 of the *ALK* gene, which was first identified in colorectal cancer [53]. *SPTBN1*-*ALK* gene fusion is a potential biomarker of cancer refractory to therapy and of a relatively poor prognosis. However, the novel *SPTBN1*-*ALK* fusion gene may become a potential target for anti-tumor therapy [54].

#### 2.2.4. Cell Cycle Gene Alterations

Cell cycle gene alterations include cyclin D (*CCND*) amplification, cyclin E1 (*CCNE1*) amplification, cyclin-dependent kinase 4/6 (*CDK4/6*) amplification, and *CDKN2A E27fs* [50,52].

#### 2.2.5. MAPK/PI3K alterations

MAPK/PI3K alterations include v-raf murine sarcoma viral oncogene homolog B1 (*BRAF*)^V600E^ mutation [55], phosphoinositide-3-kinase P110α catalytic subunit (*PIK3CA*)^E545K^ mutation [50], and Kirsten rat sarcoma viral oncogene homolog (*KRAS*)^G12D^ mutation [50]. These MAPK/PI3K alterations will be described in detail in Section 2.4.1, Section 2.4.2 and Section 2.4.3.

#### 2.2.6. Others

Other mechanisms of acquired resistance to osimertinib include wild-type *EGFR* amplification [33], loss of T790M and other activated mutation [33], fibroblast growth factor receptor (*FGFR*) amplification [33], mesenchymal-epithelial transition factor (*MET*) amplification [56], human epidermal growth factor receptor 2 (HER2) [56], and SCLC transformation [33]. Resistance mechanisms are heterogeneous, and each number is relatively small. An in-vitro study demonstrated that the combination of rociletinib—a third-generation EGFR-TKI—or afatinib with cetuximab has an effect on wild-type *EGFR* amplification [57]. Loss of T790M was associated with a slightly shorter median survival. Use of cytotoxic agents is a therapeutic option in patients with loss of T790M without other mutation, and SCLC transformation.

### 2.3. Activation of Alternative Pathways

#### 2.3.1. *MET* Gene Amplification

The most common alternative pathway is *MET* amplification, accounting for 5% to 10% of cases of acquired resistance to EGFR-TKIs [16,58]. The *MET* gene encodes the MET tyrosine kinase receptor. MET is activated by the hepatocyte growth factor (HGF) and potentiates survival through activation of ERBB3/phosphoinositol-3-kinase (PI3K)/protein kinase B (AKT) signaling [58,59,60]. *MET* amplification induces autophosphorylation of MET protein and associates with ERBB3 which activates the PI3K/AKT pathway, leading to the development of resistance to EGFR-TKIs [58]. Crizotinib is a kinase inhibitor with multiple targets (including MET). Two case reports have reported that crizotinib was effective against *EGFR*-mutated NSCLC harboring *MET* amplification [61,62]. The novel potent and selective MET inhibitor capmatinib (INC280) has demonstrated preclinical activity in combination with gefitinib in *EGFR*-mutant, *MET*-amplified/overexpressing models of acquired resistance to EGFR-TKIs. A phase Ib/II study investigating the safety and efficacy of capmatinib plus gefitinib in patients with *EGFR*-mutated and *MET*-dysregulated (amplified/overexpressing) NSCLC revealed an ORR of 47% in patients with a *MET* gene copy number ≥6 with tolerable toxicity [63].

#### 2.3.2. HGF Overexpression

HGF overexpression is frequently observed in 61% of patients with acquired resistance to EGFR-TKIs [64]. A high level of HGF in the serum is a poor prognostic factor in NSCLC patients treated with first- and second-generation EGFR-TKIs [65,66,67]. The HGF activates MET and induces resistance to EGFR-TKIs via the PI3K/AKT pathway [68]. In contrast to *MET* amplification, HGF overexpression induces resistance to EGFR-TKIs by activating the MET/PI3K/AKT pathway without the involvement of human epidermal growth factor receptor type 3 (HER3, also known as ERBB3) [68]. In a preclinical study, the combination of onartuzumab—a monovalent antibody to MET—with erlotinib blocks the HGF-induced activation of MET [69].

#### 2.3.3. Insulin-Like Growth Factor (IGF) Upregulation

The IGF receptor 1 (IGF1R) is a transmembrane heterotetrameric protein playing a role in the promotion of oncogenic transformation, growth, and survival of cancer cells [70]. IGF1R activates two signal transduction pathways, namely the RAS/RAF/MEK/ERK and PI3K/AKT pathways. IGF1R plays a role in the development of resistance to gefitinib in the absence of the *EGFR* T790M mutation [71]. In addition, IGF1R is associated with the development of resistance to irreversible second-generation EGFR-TKIs in *EGFR* T790M-mutated lung cancer cell lines [71]. The small molecule IGF1R inhibitor AG-1024 or the monoclonal anti-IGF1/2 blocking antibody BI 836845 has been reported to restore sensitivity to treatment with third-generation EGFR-TKIs [72].

#### 2.3.4. HER2 Amplification

HER2/ERBB2 is a member of the ERBB family and is unable to form homodimers because HER2 lack its specific ligand which has a crucial role in dimerization [73]. Therefore, HER2 is present either solely as monomer or heterodimer with other ligand-bound family members including EGFR [73,74]. HER2 and EGFR indirectly activate PI3K, and amplification of HER2 is associated with the development of acquired resistance to EGFR-TKI occurring in 12% of active *EGFR*-mutated NSCLC patients who develop resistance [75]. The combination of afatinib and cetuximab inhibits HER2 phosphorylation and delays the development of resistance compared with single-agent treatment (erlotinib or afatinib) in vitro and in vivo [75,76]. Trastuzumab—a human monoclonal antibody that interferes with the HER2 receptor—exerted a modest effect in lung cancer patients with *HER2* amplification and HER2 overexpression [77,78]. A phase II trial involving *EGFR*-mutated NSCLC patients with HER2 activation showed an ORR of 41% and a median duration of response of 9 months [79].

#### 2.3.5. Growth Arrest-Specific 6 (GAS6)/ Anexelekto (AXL) Activation

The GAS6 is a ligand of AXL which is receptor tyrosine kinase. The activation of the *AXL* by GAS6 leads to activation of the MEK/ERK and PI3K/AKT pathways and acquired resistance to EGFR-TKIs [80]. Furthermore, GAS6/AXL activation is often accompanied by EMT [80]. The VEGFR/MET/AXL inhibitor foretinib restored partial sensitivity of EGFR-TKI-resistant cells [27]. Furthermore, multiple AXL inhibitors such as multitarget kinase inhibitors (S49076, cabozantinib, ASLAM002, MGCD265, and MGCD516) and a specific AXL inhibitor (BGB324) are currently being developed for the treatment of solid tumors including NSCLC [81].

### 2.4. Activation of Downstream Targets

Schema of downstream signaling of EGFR is shown in Figure 3.

#### 2.4.1. *KRAS* Mutation

Previously, the *KRAS* mutation and *EGFR* were considered to be exclusive genes. Indeed, *KRAS* and *EGFR* were only found in <2% of NSCLC patients [82,83,84,85]. Moreover, the *KRAS* mutation was observed only in a mouse model, unlike in tissue samples obtained from patients with acquired resistance to first- and second-generation EGFR-TKIs [15,23,86,87]. Recently, analysis of circulating tumor DNA revealed the emergence of distinct *KRAS* activating mutations (G12A, Q61H, and A146T) in three of 43 *EGFR*-mutant NSCLC patients with acquired activating mutations [41]. The RAS inhibitor was discovered through in-silico screening. This inhibitor suppresses the metastasis of colorectal cancer cell lines via the downregulation of lysyl oxidase [88,89]. However, the effect of RAS inhibitor on EGFR-acquired resistance remains unclear.

#### 2.4.2. *BRAF* Mutation

*BRAF* mutations (G469A and V600E) were found in two of 195 (1%) patients with acquired resistance to EGFR-TKIs [87]. The combination of encorafenib (LGX818)—a *BRAF*^V600E^ inhibitor—with osimertinib suppresses the colony formation of cells, which were collected and cultured at the time of progression in malignant pleural effusion of patients treated with osimertinib [55].

#### 2.4.3. *PIK3CA* Mutation

*PIK3CA* encodes p110α and is one of the class I PI3K isoforms [90]. It was found in one of 37 (3%) patients with acquired resistance to EGFR-TKIs [15]. The dual inhibitor of PI3K/mTOR NVP-BEZ235 effectively inhibits the growth of gefitinib-resistant NSCLC cells by down-regulating PI3K/AKT/mTOR phosphorylation [91]. In a phase I clinical trial, the PI3K inhibitor temporarily decreased the size of the tumor in a patient with *EGFR* L858R/T790M/PIK3CA mutation after progression during treatment with erlotinib. However, the tumor relapsed within 1 month [92]. A phase II study showed that the combination of the AKT inhibitor MK-2206 with erlotinib in patients with advanced *EGFR*-mutated NSCLC who previously progressed during treatment with erlotinib showed an ORR of 9% and a PFS of 4.4 months [93].

#### 2.4.4. Phosphatase and Tensin Homolog (*PTEN*) Deletion

*PTEN* at chromosome 10q23.3 is a tumor suppressor gene with sequence homology to protein tyrosine phosphatases. The *PTEN* gene is inactivated in several types of human tumors by mutation, homozygous deletion, promoter methylation, and translational modification [94]. It negatively regulates the PI3K/AKT signaling pathway [95]. The loss of *PTEN* function increases the level of phosphatidylinositol-(3,4,5)-triphosphate (PIP-3; a product of PI3K) and leads to AKT hyperactivation. Loss of *PTEN* has been shown to be involved in the development of resistance to EGFR-TKIs in certain tumor cell lines [96,97,98]. Recently, peroxisome proliferator-activated receptor gamma (PPARγ) agonist drugs such as rosiglitazone sensitize *PTEN*-deficient resistant human lung cancer cells to EGFR tyrosine kinase inhibitors by inducing autophagy [99].

#### 2.4.5. *NF-1* Deletion

*NF-1* is a tumor suppressor gene encoding a neurofibromin, namely GTPase-activating protein that negatively regulates p21-RAS signaling [100]. *NF-1* exerts a reverse effect on RAS by increasing the guanosine triphosphate (GTP) hydrolysis rate. Hence, its function as a tumor suppressor is believed to occur by constraining RAS activity in the normal cell. *NF-1* deletion activates the MAPK pathway and was associated with the development of primary and acquired resistance of lung adenocarcinomas to EGFR TKIs in patients [101]. A MAP–ERK kinase (MEK) inhibitor restored sensitivity to erlotinib in lung cancer which was reduced *NF-1* expression in vitro and in vivo [101]. Indeed, patients with plexiform neurofibromas, whose tumors harboring *NF1* mutation, have shown response to this strategy of downstream inhibition [102].

#### 2.4.6. v-crk Sarcoma Virus CT10 Oncogene Homolog (Avian)-Like (*CRKL*) Amplification

*CRKL* is an oncogene encoding an adaptor protein that participates in signal transduction of the RAS/RAF/MAPK pathway [103]. *CRKL* amplification was detected in EGFR-TKI-treated lung adenocarcinoma patients [103]. *CRKL*-specific short hairpin RNA (shRNA) suppressed the tumorigenic growth of NSCLC cells harboring the *CRKL* amplification [103].

### 2.5. Histologic Transformation

#### 2.5.1. EMT

EMT is critical steps of morphogenesis by interconverting epithelial cell types into cells with mesenchymal attributes [104]. Molecular changes, including the loss of cell junction proteins (e.g., E-cadherin and claudins) or up-regulation of mesenchymal markers (e.g., vimentin and fibronectin), are associated with EMT [105]. Transforming growth factor (TGF)-β, Wnt, and Notch signals from the microenvironment are also associated with the induction and maintenance of EMT [104,106,107,108]. EMT was detected in 1% to 2% of patients with resistance to EGFR-TKIs [15,80,109]. Inappropriate induction of EMT in tumor cells is associated with tumor invasion, metastasis, drug resistance, and stem cell property [110,111]. Some cytotoxic agents (i.e., cisplatin, gemcitabine, etoposide, and vinorelbine) have demonstrated an antitumor effect in cells with a mesenchymal phenotype, whereas other cytotoxic agents (i.e., docetaxel and pemetrexed) are not effective in these cells [18].

#### 2.5.2. SCLC Transformation

Transformation from *EGFR*-mutated adenocarcinoma to SCLC occurs in 5% of patients with acquired resistance to EGFR-TKIs [15]. Recent studies revealed that SCLC transformation is associated with inactivation of Rb and p53 [112,113]. Chemotherapy with etoposide and carboplatin was reported to be effective in two cases of SCLC transformation from *EGFR*-mutated adenocarcinoma [114].

## 3. Primary Resistance

### 3.1. HGF Overexpression

In a study, 29% of cases showed high resistance to HGF (primary resistance) and did not show marked response to EGFR-TKIs, despite the presence of an EGFR mutation [64]. In addition, the HGF reduced the susceptibility to irreversible EGFR-TKIs in H1975 cells harboring both the L858R activating mutation and T790M mutation in EGFR [115].

### 3.2. Decrease of BIM

Bcl-2-like protein 11 (BIM) is a member of the Bcl-2 protein family and a crucial mediator of apoptosis. *EGFR*-mutated lung cancer cells with low expression levels of BIM exhibit low sensitivity to treatment with EGFR-TKIs [116]. A decrease in *BIM* polymorphism, which was thought to be primary resistance to EGFR-TKI, was detected in 12% of East Asians [18]. Use of proapoptotic BH3 mimetics may be effective for the decrease of BIM.

### 3.3. Others

*EGFR* exon 20 insertions [117,118], the *PIK3CA* mutation [119], and *PTEN* deletion contribute to inhibition of cell proliferation and survival or suppression of apoptosis, which lead to primary resistance to EGFR-TKIs [120]. The majority of patients with advanced lung adenocarcinomas harboring *EGFR* exon 20 insertions do not respond to treatment with EGFR-TKIs. However, cytotoxic chemotherapy in these patients offers a similar overall survival to that observed in patients with a sensitizing *EGFR* mutation [121].

## 4. Summary

We reviewed the mechanisms of acquired and primary resistance to EGFR-TKIs. The development of appropriate treatment strategies according to the different mechanisms of resistance is warranted. Overcoming resistance to EGFR-TKIs may prolong patient survival. Thus, further basic research on the molecular mechanisms involved in this process and clinical research on potential therapeutic options are of crucial importance.

## Figures and Tables

**Figure 1 cells-07-00212-f001:**
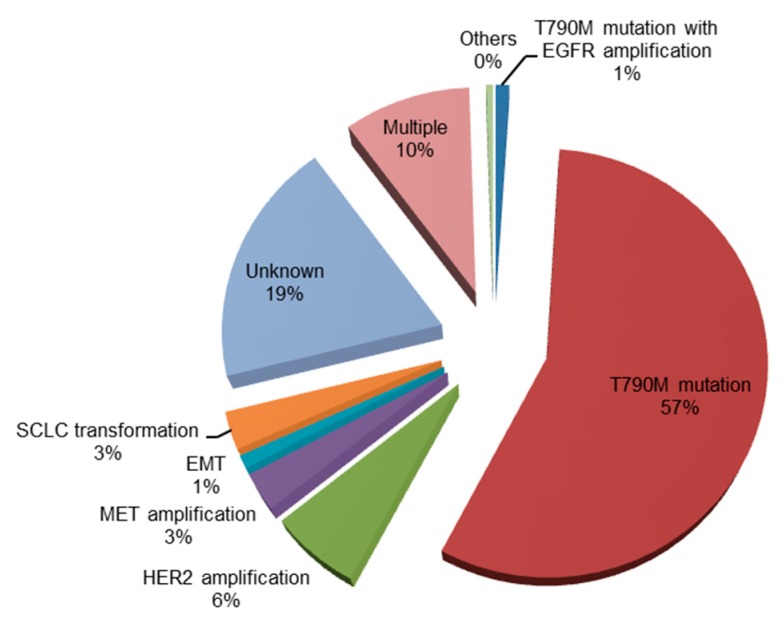
Mechanisms of acquired resistance to first-generation tyrosine kinase inhibitors (gefitinib and erlotinib) [15,16]. EGFR, epidermal growth factor receptor; HER2, human epidermal growth factor receptor 2; MET, mesenchymal–epithelial transition factor; EMT, epithelial–mesenchymal transition; SCLC, small-cell lung cancer.

**Figure 2 cells-07-00212-f002:**
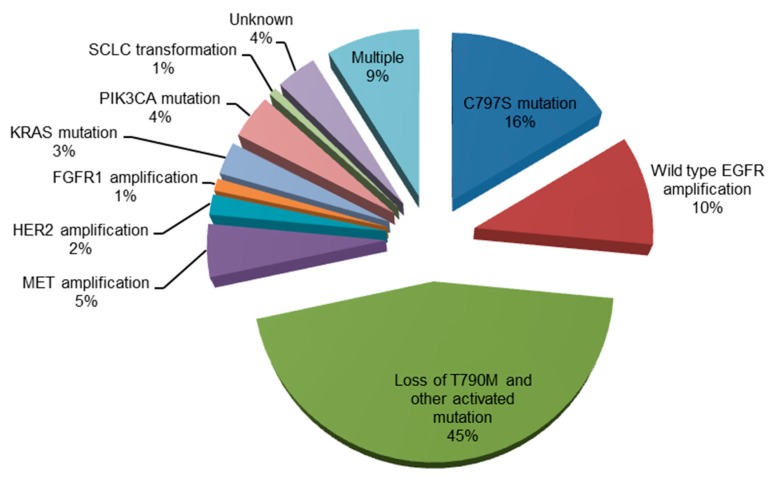
Mechanisms of acquired resistance to osimertinib [32,33,34,35,36]. EGFR, epidermal growth factor receptor; MET, mesenchymal-epithelial transition factor; HER2, human epidermal growth factor receptor 2; FGFR1, fibroblast growth factor receptor 1; KRAS, Kirsten rat sarcoma viral oncogene homolog; PIK3CA, phosphoinositide-3-kinase P110α catalytic subunit; SCLC, small-cell lung cancer.

**Figure 3 cells-07-00212-f003:**
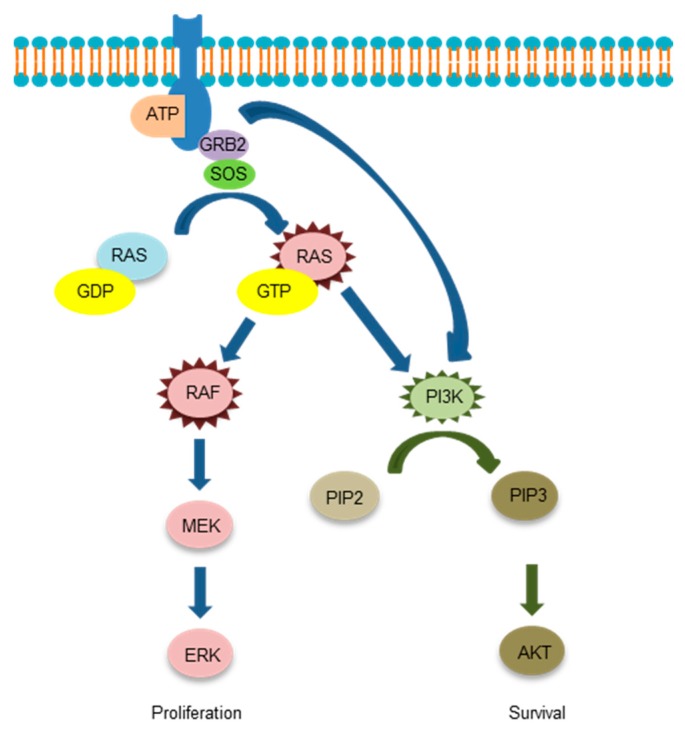
The downstream signaling of the epidermal growth factor receptor.

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
