# Peer review of "Mechanism of Resistance to Epidermal Growth Factor Receptor-Tyrosine Kinase Inhibitors and a Potential Treatment Strategy"

_cells, 2018, doi:10.3390/cells7110212_

Round 1
Reviewer 1 Report
In the study entitled "Mechanism of resistance to epidermal growth factor receptor-tyrosine kinase inhibitors and potential treatment strategy” the authors attempt to review the mechanisms of acquired and primary resistance to EGFR-TKIs. Because overcoming resistance to EGFR-TKIs may prolong patient survival, basic research on the molecular mechanisms involved in this process and clinical research on potential therapeutic options are of crucial importance.
Although the topic is interesting and the study well designed, this review does not improve the amount of recently published data. Actually, in the latest years there are numerous published papers about the same topic such as:
· Lim SM, Syn NL, Cho BC, Soo RA. Acquired resistance to EGFR targeted therapy in non-small cell lung cancer: Mechanisms and therapeutic strategies. Cancer Treat Rev. 2018 Apr; 65:1-10.
· Yang Z, Tam KY. Combination Strategies Using EGFR-TKi in NSCLC Therapy: Learning from the Gap between Pre-Clinical Results and Clinical Outcomes. Int J Biol Sci. 2018 Feb 5; 14(2):204-216.
· Dong L, Lei D, Zhang H. Clinical strategies for acquired epidermal growth factor receptor tyrosine kinase inhibitor resistance in non-small-cell lung cancer patients. Oncotarget. 2017 Aug 4;8 (38):64600-64606.
· Proto C, Lo Russo G, Corrao G, Ganzinelli M, Facchinetti F, Minari R, Tiseo M, Garassino MC. Treatment in EGFR-mutated non-small cell lung cancer: how to block the receptor and overcome resistance mechanisms. Tumori. 2017 Jul 31;103 (4):325-337.
Moreover, in some sub-chapter the authors reported only limited citations and clinical data must be improved to better understand the potential treatment strategy. For example, the authors must discuss more in details the role of PTEN or NF-1 deletion, trying to explain which effect has the deletion on clinical treatment or acquired resistance.
Author Response
Point 1: In the study entitled "Mechanism of resistance to epidermal growth factor receptor-tyrosine kinase inhibitors and potential treatment strategy” the authors attempt to review the mechanisms of acquired and primary resistance to EGFR-TKIs. Because overcoming resistance to EGFR-TKIs may prolong patient survival, basic research on the molecular mechanisms involved in this process and clinical research on potential therapeutic options are of crucial importance.
Although the topic is interesting and the study well designed, this review does not improve the amount of recently published data. Actually, in the latest years there are numerous published papers about the same topic such as:
· Lim SM, Syn NL, Cho BC, Soo RA. Acquired resistance to EGFR targeted therapy in non-small cell lung cancer: Mechanisms and therapeutic strategies. Cancer Treat Rev. 2018 Apr; 65:1-10.
· Yang Z, Tam KY. Combination Strategies Using EGFR-TKi in NSCLC Therapy: Learning from the Gap between Pre-Clinical Results and Clinical Outcomes. Int J Biol Sci. 2018 Feb 5; 14(2):204-216.
· Dong L, Lei D, Zhang H. Clinical strategies for acquired epidermal growth factor receptor tyrosine kinase inhibitor resistance in non-small-cell lung cancer patients. Oncotarget. 2017 Aug 4;8 (38):64600-64606.
· Proto C, Lo Russo G, Corrao G, Ganzinelli M, Facchinetti F, Minari R, Tiseo M, Garassino MC. Treatment in EGFR-mutated non-small cell lung cancer: how to block the receptor and overcome resistance mechanisms. Tumori. 2017 Jul 31;103 (4):325-337.
Response 1: Sincerely I would like to thank to this criticism. We added the novel acquired mechanisms which reported at latest international conference such as ESMO2018. The third-generation EGFR-TKI, osimertinib will become standard therapy for EGFR-mutated lung adenocarcinoma. Therefore, its resistance mechanism is particularly important. We believe that our review is valuable and unlike other reports in that it provides detailed information on this mechanism and provides up-to-date information.
Point 2: Moreover, in some sub-chapter the authors reported only limited citations and clinical data must be improved to better understand the potential treatment strategy. For example, the authors must discuss more in details the role of PTEN or NF-1 deletion, trying to explain which effect has the deletion on clinical treatment or acquired resistance.
Response 2: We thank this comment and added the following sentence in the role of PTEN and NF-1: “PTEN at chromosome 10q23.3 is a tumor suppressor gene”, “The PTEN gene is inactivated in several types of human tumors by mutation, homozygous deletion, promoter methylation, and translational modification [91].”, “The loss of PTEN function increases the level of phosphatidylinositol-(3,4,5)-triphosphate (PIP-3; a product of PI3K) and leads to AKT hyperactivation.”, “Recently, peroxisome proliferator-activated receptor gamma (PPARγ) agonist drugs such as rosiglitazone sensitize PTEN-deficient resistant human lung cancer cells to EGFR tyrosine kinase inhibitors by inducing autophagy [96].”, “NF-1 exerts a reverse effect on RAS by increasing the guanosine triphosphate (GTP) hydrolysis rate. Hence, its function as a tumor suppressor is believed to occur by constraining RAS activity in the normal cell. NF-1 deletion activates the MAPK pathway”, and “Indeed, patients with plexiform neurofibromas, whose tumors harboring NF1 mutation, have shown response to this strategy of downstream inhibition [99].”.
Reviewer 2 Report
The study is a mini-review on various mechanisms of resistance to EGFR Kinase inhibitors in cancer. It is interesting and well summarized. I have only minor suggestions:
Authors keep using a mixed lexicon/terminology of "gene missense mutation or gene point mutation with amino acid substitution". For example,
(1) C-->T point mutation or missense mutation in exon 21 of a certain gene represented as C455T encoding amino acid T790M....(if all of these are correct, we know that "exon" and "intron" are arranged in "gene (DNA) sequence of A,G,C,T).
(2) T790M (substitution arising from replacement of "A or G or C or T") is amino acid. With this in mind,
a. line 35: define ATP (do not abbreviate when appearing for the first time)
b. line 39: define RAS-RAF-MAPK and PI3K-AKT
c. line 46: T790M gene mutation: it is not gene. "T" and "M" are amino acids, not gene.
d. line 55: define SCLC, MET, HER2 in the figure legend for figure 1
e. line 70: define IC50
f. line 100: define KRAS in fig.2
g. line 118: define ALK
h. line 130: define MET
I. line 151: define ERBB3
j. line 196: define BRAF
Thank you
Author Response
Point 1: The study is a mini-review on various mechanisms of resistance to EGFR Kinase inhibitors in cancer. It is interesting and well summarized. I have only minor suggestions:
Authors keep using a mixed lexicon/terminology of "gene missense mutation or gene point mutation with amino acid substitution". For example,
(1) C-->T point mutation or missense mutation in exon 21 of a certain gene represented as C455T encoding amino acid T790M....(if all of these are correct, we know that "exon" and "intron" are arranged in "gene (DNA) sequence of A,G,C,T).
(2) T790M (substitution arising from replacement of "A or G or C or T") is amino acid. With this in mind,
a. line 35: define ATP (do not abbreviate when appearing for the first time)
b. line 39: define RAS-RAF-MAPK and PI3K-AKT
c. line 46: T790M gene mutation: it is not gene. "T" and "M" are amino acids, not gene.
d. line 55: define SCLC, MET, HER2 in the figure legend for figure 1
e. line 70: define IC50
f. line 100: define KRAS in fig.2
g. line 118: define ALK
h. line 130: define MET
I. line 151: define ERBB3
j. line 196: define BRAF
Response 1: Thank you for your advice. I modified the manuscript as following points.
a. line 35: I defined ATP as adenosine triphosphate.
b. line 39: I defined RAS-RAF-MAPK and PI3K-AKT as rat sarcoma (RAS)-rapidly accelerated fibrosarcoma (RAF)-mitogen-activated protein kinase (MAPK) and phosphatidylinositol 3-kinase (PI3K)- protein kinase B (PKB, also known as AKT) pathways.
c. line 46: I changed T790M gene mutation into T790M mutation.
d. line 55: I added EGFR, epidermal growth factor receptor; HER2, human epidermal growth factor receptor 2; MET, mesenchymal-epithelial transition factor; EMT, epithelial-mesenchymal transition; SCLC, small-cell lung cancer in the figure legend for figure 1.
e. line 70: I defined IC50 as half-maximal inhibitory concentration.
f. line 100: I added EGFR, epidermal growth factor receptor; MET, mesenchymal-epithelial transition factor; HER2, human epidermal growth factor receptor 2; FGFR1, fibroblast growth factor receptor 1; KRAS, Kirsten rat sarcoma viral oncogene homolog; PIK3CA, phosphoinositide-3-kinase P110α catalytic subunit; SCLC, small-cell lung cancer in fig.2.
g. line 118: I defined ALK as anaplastic lymphoma kinase.
h. line 130: I has already defined MET as mesenchymal-epithelial transition factor in the text.
I. line 151: I defined ERBB3 as human epidermal growth factor receptor type 3.
j. line 196: I defined BRAF as v-raf murine sarcoma viral oncogene homolog B1
Round 2
Reviewer 1 Report
Although the authors try to respond to Reviewer's Comments, this review is too similar to the recently published papers and the added data about osimertinib do not improve so much the information useful for the readers. Moreover, the information reported in different sub-chapter are quite simply and sometimes the literature review is not meaningful comprehensive.
Author Response
Response :
Point 1: Although the authors try to respond to Reviewer's Comments, this review is too similar to the recently published papers and the added data about osimertinib do not improve so much the information useful for the readers. Moreover, the information reported in different sub-chapter are quite simply and sometimes the literature review is not meaningful comprehensive.
Response 1: Sincerely I would like to thank to this criticism. Since this field is very active research field, we should also check the review articles and should make efforts to create a difference more carefully. We added the following sentence in Introduction “There are several outstanding reviews so far [6-9], but therapeutic drugs progress one after another and a new resistance mechanism emerges. Therefore, we would like to review focusing on the novel acquired resistance of the third generation EGFR-TKI, osimertinib based on the latest academic information.”. In addition, we organized text in order to make easier to understand as follows: “These MAPK/PI3K alterations will be described in detail in sub-chapter 2.4.1., 2.4.2. and 2.4.3.”.